# Next-Generation Examination, Diagnosis, and Personalized Medicine in Periodontal Disease

**DOI:** 10.3390/jpm12101743

**Published:** 2022-10-20

**Authors:** Takeshi Kikuchi, Jun-ichiro Hayashi, Akio Mitani

**Affiliations:** Department of Periodontology, School of Dentistry, Aichi Gakuin University, 2-11 Suemori-dori, Chikusa-ku, Nagoya 464-8651, Aichi, Japan

**Keywords:** personalized medicine, precision periodontics, biomarkers, periodontal microbiome, genetic profiling, diagnostic imaging, regenerative engineering, systemic disease, lifestyle, antimicrobial photodynamic therapy

## Abstract

Periodontal disease, a major cause of tooth loss, is an infectious disease caused by bacteria with the additional aspect of being a noncommunicable disease closely related to lifestyle. Tissue destruction based on chronic inflammation is influenced by host and environmental factors. The treatment of periodontal disease varies according to the condition of each individual patient. Although guidelines provide standardized treatment, optimization is difficult because of the wide range of treatment options and variations in the ideas and skills of the treating practitioner. The new medical concepts of “precision medicine” and “personalized medicine” can provide more predictive treatment than conventional methods by stratifying patients in detail and prescribing treatment methods accordingly. This requires a new diagnostic system that integrates information on individual patient backgrounds (biomarkers, genetics, environment, and lifestyle) with conventional medical examination information. Currently, various biomarkers and other new examination indices are being investigated, and studies on periodontal disease-related genes and the complexity of oral bacteria are underway. This review discusses the possibilities and future challenges of precision periodontics and describes the new generation of laboratory methods and advanced periodontal disease treatment approaches as the basis for this new field.

## 1. Introduction

In 2021, the 74th World Health Assembly of the World Health Organization (WHO) adopted a resolution on oral health that stated the importance of disease control and prevention related to oral health [1]. The incidence of oral disease is extremely high throughout the world, affecting many people and imposing a heavy economic burden on society. Oral diseases are not only manifested as health disparities among the poor and socially vulnerable but are also strongly associated with noncommunicable diseases (NCDs) and have been flagged as a major problem even in developed countries, leading to decreased productivity. Because many oral diseases are preventable, the WHO has set the goal of achieving improved oral health by 2030 as part of its efforts to establish universal health coverage that can provide health services to all people at a cost that they can afford to pay and to address NCDs [2]. The oral diseases that have been flagged as being associated with NCDs are primarily periodontal diseases. In developed countries with aging populations, the number of remaining teeth is increasing, while the incidence of periodontal disease is rising [3]. Periodontal disease is not only a major cause of tooth loss but also a risk factor for many NCDs, including cardiovascular disease, diabetes, lung disease, pregnancy complications, obesity, cancer, and Alzheimer’s disease [4]. Therefore, efforts to counter periodontal disease have become an important aspect of health strategies.

Current medical care revolves around evidence-based standard treatments for the purpose of assuring quality and safety. This is population-based medicine, where efficacy and risk of complications are expressed in terms of probabilities or odds, and treatments are selected accordingly. For each individual patient, however, it is not important to express effectiveness as a percentage or probability; whether it actually works or not is the concern. Even if it works for many people, it is not an optimal treatment unless it works for the individual patient. Conversely, even if a treatment is not generally recommended because of its lack of efficacy in many cases, it is still a suitable treatment when it works for the patient. This conundrum demonstrates the limitations of evidence-based medicine (EBM).

In recent years, attention has been focused on “precision medicine,” which is medical care that optimizes treatment methods for each individual patient. In the United States, President Obama announced the Precision Medicine Initiative in his State of the Union address in 2015, which led to the development of large-scale integrated research projects to collect and analyze genome information, medical information, medical samples, and life logs, towards the realization of precision medicine [5]. Although conventional medicine has also stratified patients based on a combination of test results and symptoms, precision medicine integrates genomic information to diagnose the onset of disease at the molecular and genetic levels and select treatment methods. Information on the environment and lifestyle of the patient is added to identify individual risks and characteristics with the assistance of artificial intelligence (AI), which can be used to develop strategies for prevention and treatment [6]. This is a new medical concept based on the use of a huge amount of data, and its methodologies will be developed in a variety of ways in the future. Because this medical concept was originally proposed to solve the problem that the efficacy and side effects of anticancer drugs vary greatly from patient to patient, the basic strategic model has been established in the field of cancer medicine [7]. The behavior of cancer, even in the same cell type, is influenced not only by genomic and epigenomic variations in an individual but also by the individual’s life history and environment. Therefore, for diagnosis and risk prediction, methods are used to analyze genetic information such as the cancer genes of an individual in combination with information on environmental factors [6,8]. To apply this to other diseases, basic research on disease genes is necessary, and it is important to know how much research has been accumulated to date and how many projects on data collection and database construction will be developed in the future.

In the field of periodontal disease, several new testing parameters, including various biomarkers, have been proposed, and the analysis of genes related to periodontal disease is progressing. Additionally, studies on the complexity of oral bacteria are ongoing, and the groundwork for precision medicine is steadily accumulating. This review introduces a new generation of testing methods for periodontal disease and advanced periodontal treatment, and it discusses the possibilities for precision treatment of periodontal disease.

## 2. Personalization of Periodontal Treatment

The progression of periodontal disease is complicated because of the number of affected teeth and the degree of destruction of periodontal tissue, which differs in each tooth. In addition, the adherence of bacterial plaque, defined as biofilm, and occlusal conditions that cause mechanical stress have a significant impact on the disease. Therefore, the treatment of periodontal disease has traditionally been arranged according to the individual patient’s situation. The actual adoption of the procedures and the process of occlusal reconstruction are widely selective and depend on the ideas and skills of the practitioners who carry them out. No case can proceed with the same treatment plan as for another patient. In other words, there is no single correct approach. In this sense, periodontal treatment is inherently individualized and difficult to standardize and optimize.

The new guidelines for periodontal disease classification introduced in 2018 adopt a system of evaluation based on stage classification (defined by disease severity and management complexity) and grade classification (defined by speed of progression and degree of risk factor involvement) [9]. In conventional diagnosis, the disease was classified into two categories, chronic periodontitis that progresses slowly or aggressive periodontitis that progresses rapidly, according to differences in phenotype represented by the speed of progression; the severity of disease was evaluated according to the destruction of periodontal tissue in each disease. However, there was no pathophysiological evidence to rationally explain the disease phenotype, and no valid examination criteria were found to distinguish between the two categories. Therefore, rather than defining them as separate diseases based on phenotype, they were to be defined as manifestations of disease diversity. The aim of the new definition system is to enable a multidimensional diagnostic classification that incorporates each risk factor, rather than the previous one-dimensional approach of diagnosing by the destruction of periodontal tissue [9]. There is a clear intention that the system will enable the optimization of patient management and lead to precision or personalized medicine in the future.

How might precision or personalized medicine in periodontal disease be defined? In recent years, several review articles on precision medicine in periodontal disease have been published [10,11,12]. Bartold et al. proposed a practice model for the management of periodontitis termed P4Periodontics that incorporates the concept of P4 medicine, with prediction, prevention, personalization, and participation as the four pillars of clinical patient care [10]. This model extends the new 2018 classification of periodontal disease to a treatment paradigm, in which each stage of periodontitis is associated with a holistic approach to periodontal disease management. It also states that such an approach can manipulate the aggressive and rapidly progressive course of periodontitis into a more slowly progressive course [10]. Rakic et al. note that although the driving forces behind precision periodontics are biomarkers and machine learning algorithms, at this time the practice is still in its infancy due to the lack of periodontal markers that can be used for diagnosis, and they review the utility of the biomarkers [11]. The structure of precision periodontics conceived from these discourses consists of a multidimensional diagnosis, the resulting stratification of patients into subgroups, and treatment approaches according to the characteristics of each subgroup. How periodontal disease progresses depends largely on oral bacteriological factors and the congenital and acquired biological background and lifestyle of the patient (Figure 1). Because the combination of these factors can be enormous, patient stratification should be conducted by dropping the combination into several subgroups so that the pattern and risk of disease progression can be predicted. This idea may be reflected in the new grading of periodontal disease. At present, there are three gradings, A, B, and C; however, as more information is accumulated, a more multilevel grading will become necessary. Stratification should also be based on the patient’s current stage of the progression. These stratifications will probably be implemented in the future by applying deep learning AI and other technologies based on the results of a variety of examinations. Furthermore, the actual treatment to be performed will be a combination of approaches that consider individual oral units and procedures that diagnose and determine the suitability of individual teeth. It is necessary to determine which procedure should be performed at which site and when and what approach should be taken in response to systemic factors and lifestyle factors, in accordance with the condition of the patient in each stratification. At present, the selection of treatment is based on evidence according to practice guidelines; however, in the future, AI will be needed to select highly predictive treatment using more detailed and voluminous data. As a basis for this selection, databases to which reference will be made include not only systematic reviews with high levels of evidence but also individual case reports.

## 3. Examination and Diagnosis for Precision Periodontics

### 3.1. Biomarkers for Diagnosis and Prognosis

Diagnoses of susceptibility to periodontal disease have been extensively studied. Currently, the treatment of periodontal disease is generally handled by a periodontist after the destruction of periodontal tissue is recognized [13]. Bleeding on probing (BOP), which is confirmed by inserting a periodontal probe into the base of the gingival sulcus or periodontal pocket, is the classic and still the most reliable method of examination for monitoring the health or progression of periodontal disease. Repeated positive BOP is predictive of future attachment loss in 30% of cases, while repeated negative BOP is indicative of healthy periodontal tissue in nearly 100% of cases [14]. The identification of biomarkers that can predict prognosis with high probability is desired for the early detection and control of periodontal disease. Traditionally, proteomic analysis has been performed using saliva [15,16,17,18], gingival crevicular fluid [19,20,21], pocket-associated tissue [22], and serum [23] from periodontal disease patients to comprehensively analyze disease-related biomarkers. Proteome analyses are also being conducted on the effects of diabetes [24], rheumatoid arthritis [25], and smoking [26], which are concomitant with periodontitis, on biomarker expression. In these papers, several biomarkers have been proposed as being associated with periodontal disease.

The diagnostic validity of a biomarker is confirmed by its sensitivity (whether the disease is truly present) and its specificity (whether the disease is truly absent). The most well-studied futuristic biomarkers are factors related to the immune response, such as cytokines and host-derived enzymes [27]. Interleukin-1beta (IL-1β) is a representative cytokine produced by macrophages and fibroblasts upon lipopolysaccharide stimulation and is deeply involved in alveolar bone resorption. The sensitivity of IL-1β ranged from 75% to 88% and specificity from 52% to 100% [28,29,30,31]. IL-6 is also produced by the inflammatory state of macrophages and is deeply involved in bone destruction. The sensitivity of IL-6 ranged from 53% to 80% and specificity from 48% to 87% [28,29,31]. Although tumor necrosis factor (TNF)-α appears to be deeply involved in disease onset and progression in experimental animal models, these percentages are relatively low. Macrophage inflammatory protein (MIP)-1α is a chemotactic cytokine expressed by numerous cells, including macrophages and neutrophils, and it plays a role in promoting chemotaxis and vascular endothelial mobility. Studies investigating MIP-1 have reported values of 95% and 66% for sensitivity, and 97% and 68% for specificity [29,32]. Matrix metalloproteinase (MMP)-8 is an enzyme released primarily from neutrophils that has been extensively investigated as a biomarker and has been implicated in periodontal disease staging and grading [33,34]. The sensitivity of MMP-8 ranged from 65% to 87% and specificity from 48% to 87% [28,29,31,35]. Additionally, interferon (IFN)-γ, MMP-9, MMP-12, receptor activator of the nuclear factor-Κb ligand (RANKL), osteoprotegerin (OPG), alkaline phosphatase (ALP), cathepsin B, hemoglobin, salivary neuropeptides, and oxidative stress-related and terminal glycation end products are also being investigated as biomarkers of periodontal disease, although their usefulness is uncertain [36].

### 3.2. Evaluation of the Periodontal Microbiome

The oral cavity is constantly exposed to miscellaneous bacteria from the external environment. Among them, approximately 700 strains of bacteria are thought to be established as indigenous to the oral cavity, and the proportions of these strains differ greatly from one individual to another [37]. Even in the same individual, the balance varies greatly depending on the condition of oral cleaning and the disease state [38]. Early attempts to link the composition of the oral bacterial flora to periodontitis were made by DNA–DNA hybridization. The most significant result was the definition of red complex bacteria by Socransky and colleagues [39]. Red complex bacteria, namely *Porphyromonas gingivalis*, *Treponema denticola*, and *Tannerella forsythia*, are frequently detected simultaneously in foci of chronic periodontitis, suggesting that these bacteria maintain a symbiotic relationship and are involved in the pathogenesis of periodontitis.

These bacteria account for only a small percentage of the total bacterial flora; other more highly prevalent bacterial species are present, and the low levels of red complex bacterial virulence factors are quantitatively insufficient to fully explain the pathogenesis of periodontitis. Hajishengallis et al. proposed the keystone hypothesis and the polymicrobial synergy and dysbiosis (PSD) model that low abundance bacterial species can have such a large impact on the host as to trigger periodontitis [40,41]. The hypothesis infers that small amounts of keystone bacteria act on the host immune system to induce host modulation, thereby altering the commensal microbiota quantitatively and qualitatively, leading to an increase in the dysbiotic community and finally disrupting tissue homeostasis. This means that the virulence factors of red complex bacteria such as *P. gingivalis* may enhance the effects on tissues by modulating the entire bacterial community, rather than directly inducing inflammation. This idea now has broad acceptance and has become a popular concept of pathogenicity of the periodontal flora.

Attempts are underway to understand the shifts in the bacterial flora of the plaque biofilm in a holistic and comprehensive approach [42]. In this process, the bacterial flora are described as a microbiome, not only as a community of bacteria but also as a concept that includes genetic material in relation to the ecology and environment [43,44]. 16SrRNA analysis has been greatly advanced by next-generation sequencing technology, enabling large-scale operations such as the Human Microbiome Project [45]. Much has also been learned about the oral microbiota [46,47].

Although the oral microbiome is second to the gut in the abundance of bacterial species, 95% of individuals have a common core species of more than a dozen taxa (including Fusobacterium, Streptococcus, and Veillonella), the proportions of which vary widely between individuals [48]. Boutin et al. used 16SrRNA cluster analysis to cluster several ecotypes of plaque microbiome and associated each with the prevalence and severity of periodontal disease [49]. Such typing could provide a basis for patient stratification by microbiological risk profile. Studies have also attempted to compare the bacterial flora in patients with periodontitis before and after treatment using next-generation sequencing analysis of 16SrRNA [50], but the results of these studies were inconsistent. While some reports indicated that the bacteriological diversity(α-diversity) of individual samples decreased with treatment and that the composition of the bacterial flora was widely divergent before and after treatment (β-diversity) [51,52,53], others reported contradictory results [54,55].This suggests that the impact of periodontal treatment on the oral microbiome may be more complex than commonly thought and cannot be told as a simple story of a change from bad flora to good flora.

Microbiome typing with 16SrRNA is a method of describing the characteristics of individual plaques in terms of their bacterial species. Metagenomic analysis, however, is an analytical method that investigates what bacterial genes are contained in that plaque and can be profiled by gene function. Wang et al. sequenced 16 metagenomic samples and analyzed their functional profiles [56]. Many functional genes and metabolic pathways, including bacterial chemotaxis, glycan biosynthesis, and lipopolysaccharide synthesis, were highly enriched in the periodontal microbiome. In a site-specific functional profiling of the oral microbiome by Dabdoub et al., energy utilization was specialized via oxidative pathways in the healthy metagenomic sample, whereas fermentation and methanogenesis were the main energy transfer mechanisms in periodontitis [57]. In addition, enhanced functions of fermentation, antibiotic resistance, detoxification stress, adhesion, invasion, intracellular tolerance, proteolysis, and quorum sensing were observed in periodontitis. Shi et al. reported that the microbiome in periodontal disease had significantly enriched genes related to carbohydrate metabolism, translation, replication, and repair compared with healthy controls; conversely, genes related to amino acid metabolism, energy metabolism, and membrane transport were significantly reduced [58].

Metagenomic analysis has made it possible to analyze the species and functional characteristics of bacterial flora and to cluster patient samples. Transcriptome and metabolome analyses are beginning to be performed, and it is imagined that more active aspects of the bacterial flora will be revealed. Technological advances will yield a great deal of information, but the key to the future will be what portion of that information should be used to stratify patients for precision medicine and what should not.

### 3.3. Specific Genetic Profile and Epigenetics

Susceptibility to periodontal disease is understood as a patient’s disposition that results from genetic factors [59]; this has led to the exploration of genes responsible for periodontal disease. In early studies, candidate genes involved in host immunity, inflammation, and tissue remodeling were selected as targets, and their single nucleotide polymorphisms (SNPs) were associated with periodontitis. A number of candidate gene studies showed that IL1A, IL1B, IL4, IL6, IL10, TNFA, FcγR, VDR, TLR2, TLR4, and MMP1 [60,61,62,63,64,65,66,67,68,69] were the main genes associated with periodontal disease. In particular, the IL-1 gene cluster has been analyzed since the early days of studies on periodontal genetics, and SNPs in the IL-1 genes were proposed to be used for genotyping periodontal patients [70,71,72,73].

In addition to investigations of genetic polymorphisms as risk markers for the onset and progression of periodontitis, many candidate gene studies have attempted to distinguish chronic periodontitis (CP) and aggressive periodontitis (AP) by their associated polymorphisms. A meta-analysis by Nikolopoulos et al. found that in Caucasians, polymorphisms of IL1A -889T/C and IL1B 3953/4 C/T were associated with CP, but they found no polymorphisms associated with AP [74]. For the TNF-α polymorphism, TNFA-308G/A was found to be associated with CP and AP, but no polymorphism distinguishing the two was found [64,75]. In a report using the polymorphisms of a long noncoding RNA, ANRIL (antisense noncoding RNA in the INK4 locus), as a marker for the diagnosis of CP and AP, the rs1333048 polymorphism was strongly associated with the new classification grade C (equivalent to the old classification AP) [76].

There have thus been many attempts to distinguish CP and AP genetically, but there is a reproducibility problem. Therefore, even if a polymorphism is significant as a risk marker in association with a disease type, it cannot be used as a diagnostic marker to identify individual disease types. This is a limitation of genotyping based on candidate gene studies.

In recent years, the genome-wide association study (GWAS) has become the mainstream method of polymorphism analysis [77], and many GWASs on periodontal disease have been reported [78,79]. In the first attempt, Schaefer et al. reported that SNP rs1537415 in glycosyltransferase gene GLT6D1 showed a strong correlation with AP [79]. Recently, many reports have shown correlations with polymorphisms in SIGLEC5 (sialic acid binding Ig-like lectin 5) [80,81]. GWASs for periodontal disease have been conducted not only in Caucasians but also in many other ethnic groups [82,83,84,85]. However, many reports have shown that GWAS did not detect any polymorphisms associated with periodontal disease [86,87].

Attempts to genotype CP and AP, whether by candidate gene studies or GWAS, initially classify subjects into groups according to clinical diagnosis, but the problem has been that the definition of the diagnosis is ambiguous and cannot be precisely distinguished. Offenbacher et al. performed a principal component analysis using bacteriological, immunological, and other data in addition to clinical diagnosis and divided the subjects into six clusters, defining each group as periodontal complex traits (PCTs) [88]. Although this was not a clinical classification but rather a mathematical stratification based on the degree of approximation calculated from the influence of multiple factors, each PCT could be characterized; for example, as a population having a high correlation and positive loading with all pathogens (PCT1), a population in which *Aggregatibacter actinomycetemcomitans* was detected (PCT3), and a population with a high proportion of *Porphyromonas gingivalis* (PCT5). When they attempted a GWAS on that group, they identified polymorphisms associated with PCT1, PCT3, and PCT5 [88].

This suggests that patient stratification, which is important for finding diagnostic genetic markers, is difficult based on the clinical phenotype of the disease alone. With current and ongoing attempts to find genetic diagnostic markers, it is not yet defined how stratification should be performed. In any case, stratification based on mathematical calculations is necessary to reflect the combined effects of multiple factors.

Epigenetic modifications affect the level of gene expression and are closely linked to disease progression [89]. Epigenetic analysis in periodontitis is also underway, with potential applications in disease risk stratification and targeted therapy [90,91]. In the early years, the methylation of DNA in the promoter region of the target gene was primarily analyzed. Starting with a report that the methylation of CpG sites in the promoter of IFNγ was significantly higher in the gingiva of periodontitis patients [92], methylation of TLR2 promoter [93], TNF-α promoter [94], and others were reported to be associated with periodontitis. In addition to targeted gene studies, whole genome studies are underway. Kim et al. detected 43,631 differentially methylated positions between periodontitis and healthy samples [95]. This indicates that the transcriptional regulation of genes is affected by methylation across a much wider range, resulting in increased or decreased gene expression in periodontitis. Genome-wide CpG methylation analysis of inflammatory and noninflammatory gingiva samples from periodontal patients revealed strong differences in the genes involved in wound healing, cell adhesion, and innate immune responses [96].

While SNPs present an inborn and immutable genomic profile, epigenomic profiles represent individual characteristics and variability depending on the state of tissue inflammation, which complicates the interpretation of their meaning as markers.

### 3.4. Diagnostic Imaging

Systems that collect individual patient medical information, such as patients’ own heart rate and blood glucose levels measured over time and stored in a database via a network, are becoming popular. However, while the use of such health information shows benefits to patients [97,98], it still poses challenges to economic sustainability. Currently, medical artificial intelligence (AI) for recommending medical examinations is expected to be introduced as soon as possible. For example, a simple system that takes intraoral photos with a smartphone and allows the AI to determine the status of periodontal disease (e.g., estimation of PPD) is eagerly awaited.

Meanwhile, the application of AI in alveolar bone radiology has been gaining traction in recent years [99]. The method of Lin et al. can effectively identify areas of alveolar bone loss in periodontitis radiographs, suggesting that it is useful for dentists to assess the degree of bone loss in patients with periodontitis [100]. In another report, Lin et al. suggested that an automated measurement system can effectively estimate the degree of horizontal alveolar bone loss in periodontitis radiographs [101]. A recent report suggested that a deep learning model could enable reliable radiographic bone loss measurements and image-based 2018 periodontal disease stage and grade classification [102]. A computer-assisted detection system based on a deep convolutional neural network (CNN) algorithm has been developed, showing relatively high accuracy in diagnosing periodontally compromised teeth (PCT) and predicting tooth extraction [103]. A method using fuzzy aggregation operators with the aim of identifying the diseases related to teeth and periodontal tissue from dental x-ray images shows the possibility of diagnosing root fracture, dental caries, alveolar bone resorption, etc. [104]. AI-based automated diagnosis based on imaging will play a major role in improving the probability and accuracy of diagnosis, considering the need for imaging in routine clinical practice.

## 4. Next-Generation Procedure in Periodontal Disease

### 4.1. Regenerative Engineering

In general, vertical intrabony defects, which are indications for periodontal tissue regenerative therapy, progress more rapidly than horizontal bone defects [105]. The ideal approach to these bone defects is crucial for the longevity of the tooth. Induction of three cell types, cementoblasts, periodontal ligament (PDL) cells, and osteoblasts, is necessary to govern periodontal tissue regeneration, and clinically applied approaches to periodontal tissue regeneration to date have focused on proliferating and differentiating these cells [106]. Guided tissue regeneration (GTR) [107], which specializes in securing the scaffold for regeneration; enamel matrix derivative (EMD) to induce cementum formation [108]; fibroblast growth factor (FGF)-2 to induce angiogenesis and the proliferation of periodontal tissue component cells [109]; platelet-derived growth factor (PDGF) to promote the healing of wounds via chemotaxis and mitogenesis [110]; and bone grafting mainly for alveolar bone regeneration are applied [106]. However, current periodontal tissue regeneration therapy is limited in the amount of tissue regenerated, and its indications are restricted to some bone defect conditions. Fortunately, a growing body of evidence demonstrates the effectiveness of new intervention methods that promote periodontal tissue regeneration.

Many comprehensive reviews focus on periodontal tissue regeneration using mesenchymal stem cells in animal models [111,112,113,114]. These results suggest that stem cell transplantation may induce periodontal tissue regeneration. Clinical studies in humans have suggested that stem cell transplantation, in combination with the use of bone substitutes and collagen, is safe and clinically useful [36,115,116,117]. Clinical and radiographic outcomes were improved in a 12-patient study of autologous PDL-derived cell sheets in combination with β-tricalcium phosphate (β-TCP) bone replacement material [118]. However, in these studies, improvements in clinical parameters are not always statistically significant. Further clinical studies are needed, along with long-term observation. The results of a comparative study of cell sheet transplantation in a canine severe defect model (one-wall intrabony defect) using cells derived from three types of mesenchymal tissue (PDL, periosteum [119], and bone marrow [120]) showed differences in the quality of periodontal tissue regeneration depending on the cell source [121]. The regeneration of periodontal ligament tissue by the delivery of adipose-derived stem cells [122] and dental pulp stem cells [123] to defects has also been reported. Conditioned medium obtained from stem cells induces periodontal tissue regeneration and suppresses inflammatory cytokine production in animal models [124]. Exosomes released from stem cells have also been suggested to induce periodontal tissue regeneration in animal models [125,126]. These cell-free procedures have the potential to reduce the risk of tumor formation and immune rejection and provide improved convenience in terms of storage and treatment costs.

Additive manufacturing is a technology that automatically manufactures three-dimensional structures by continuously depositing layer by layer the materials directed by computer-aided design (CAD) software [127]. Originally, the periodontium was represented as a typical layer-by-layer structure consisting of cementum, PDL, and alveolar bone [128,129]. Previous attempts have been made to mimic the actual periodontal tissue structure, for example by autografting polyglycolic acid-woven tri-layered PDL cell sheets onto root surfaces with three-walled bony defects filled with porous β-TCP in a dog model [130]. Additive manufacturing represents a paradigm shift in manufacturing at the level of the individual patient and has great potential to transform the field of personalized dentistry and improve regenerative outcomes in patient care. Vaquette et al. demonstrated that the combination of multiple PDL cell sheets and a biphasic scaffold allows the simultaneous delivery of the cells necessary for rat in vivo regeneration of alveolar bone, periodontal ligament, and cementum [131]. In the rat periodontal tissue defect model, the use of PDGF and bone morphogenetic proteins (BMP)-7 in combination with micropatterned scaffolds containing PDL cells showed better periodontal tissue regeneration [132]. Rasperini et al. reported a human case of treatment of a large periodontal bone defect using a 3D printed bioabsorbable patient-specific polymer scaffold and a signaling growth factor [133]. These results, while confirming safety, are not always favorable, and further improvements are needed.

### 4.2. Approach for Systemic Disorder in Periodontitis

A number of systemic diseases have been reported to affect the pathogenesis of periodontal disease [134]. Diabetes, in particular, is thought to have a significant impact on the pathogenesis of periodontal disease [135]. The impact of diabetes on periodontal disease may be mediated through diverse mechanisms [136]. Identifying the specific impact of diabetes treatment on improving periodontal disease would be beneficial in the treatment of patients with multiple comorbidities. The hormone glucose-dependent insulinotropic polypeptide (GIP), which is elevated by dipeptidyl peptidase-4 (DPP-4) inhibitors (incretin-related drugs) for the treatment of type 2 diabetes, has been shown to suppress inflammation in experimental periodontitis [137]. In contrast, glucagon-like peptide-1 (GLP-1) receptor agonists, a treatment for type 2 diabetes, have been shown to reduce alveolar bone resorption caused by experimental periodontitis [138]. These have shown improvement in periodontal disease when diabetes drugs are acted on in models of periodontal disease alone, suggesting that they may directly cause improvement in the periodontal disease condition when used as a diabetes drug. In practical terms, in a rat model of type 1 diabetes mellitus with experimental periodontitis, insulin treatment significantly improved periodontitis without local periodontitis treatment [139]. In actual clinical practice, evaluating the impact of diabetes treatment on periodontal disease may be difficult because of a number of confounding factors, but it is important work considering the large number of patients with both diabetes and periodontal disease.

Rheumatoid arthritis (RA) patients have a higher incidence of periodontal disease than healthy individuals [140]. RA and periodontal disease share many similarities in etiology and pathogenesis, and both are characterized by excessive production of inflammatory cytokines such as TNF and IL-6 [141]. In a study examining the effect of the TNF inhibitor adalimumab on periodontal status in RA patients, gingival index (GI), BOP, and probing pocket depth (PPD) were significantly improved [142]. Similarly, in a study that examined the effect of the IL-6 inhibitor tocilizumab on periodontal status in RA patients, GI, BOP, PPD, and clinical attachment level (CAL) were significantly improved [143]. These biological products have demonstrated effective action on the pathology of RA, and if they also show action on periodontal pathology, it would be beneficial for patients suffering from both diseases.

Not only the aforementioned systemic diseases but also diseases that share a broad etiology with periodontal disease should be expected to improve through these approaches. The elucidation of the relationship between periodontal disease and systemic disease, especially the mechanism of action, will allow for new personalized treatment options to be considered in the future.

### 4.3. Lifestyle and Behavioral Approach for Periodontitis

In recent years, periodontitis has been considered a noninfectious disease and is closely related to other noninfectious diseases. Defining how lifestyle and behavior affect and alter the clinical manifestations of periodontal disease at the individual level will become essential with the development of precision medicine for periodontal disease [144]. Habits that influence periodontal disease include poor oral hygiene, smoking, stress, and unhealthy diets. However, it is not easy to change and improve behavior, and health care professionals must actively work to encourage improvement using methods based on behavioral science [145].

Adequate plaque control cannot be established by patients themselves without any support. In a systematic review, Newton et al. [146] attempted to determine the associations between adherence to oral hygiene instruction and psychological constructs in adult periodontal patients and concluded that goal setting, self-monitoring, and planning are effective interventions for improving oral hygiene-related behaviors in periodontal patients. As viewed in individual studies, Suresh et al. evaluated the effectiveness of an action-controlled behavioral intervention (diary) on dental flossing adherence in 73 patients with periodontal disease [147]. They reported that the intervention improved self-reported flossing frequency, dental plaque, and bleeding scores at 4 weeks post-intervention. Jönsson et al. used an individually tailored oral health educational program incorporating cognitive behavioral therapy to instruct 113 patients with periodontal disease in oral hygiene and evaluated its effectiveness [148]. They reported a significant improvement in gingival index and plaque index after 12 months of follow-up in the group using the program.

Smoking is a highly distinct risk factor for periodontal disease, and smoking cessation is essential for the successful treatment of periodontal disease. Behavior change methods to encourage smoking cessation were reviewed by Ramseier et al., who found that behavior change counseling for smoking cessation in dental clinics was effective in adults, suggesting a rationale for behavioral support [149]. The fact that support in the dental office plays an important role in smoking cessation is one of the major social impacts of dental care.

Psychological stress also has a significant influence on periodontal disease. When a state of psychological stress persists, diverse loads act on the metabolic, immune, and nervous systems through higher-order networks, resulting in a condition called allostatic load. This then promotes complex chronic diseases [150]. The relationship between psychological stress and periodontal disease is often seen as part of this allostatic load influence, and related studies have been conducted in recent years. Sabbah et al. reviewed this topic in detail [151]. They suggest that a complex biological indicator to be established in a series of studies will allow treatment tailored to risk level, which may be beneficial for future personalized periodontics. In addition, Spector et al. showed that stress affects periodontal disease by enhancing bacteriological factors through impaired antimicrobial defense, more pathogenic gene expression, and shifts in the oral microbiome profile with taxa composition [152].

Dietary issues have a significant relationship with periodontal disease, especially high-fat, high-glucose diets, which affect periodontal disease through obesity and diabetes [153]. In contrast, a healthy diet can help control periodontal disease. Using data from 12,110 participants in the third National Health and Nutrition Examination Survey, multivariate logistic regression analysis estimated the associations between the number of health-promoting behaviors and the prevalence of periodontitis. Persons who maintained a normal weight, performed the recommended level of exercise, and ate a high-quality diet were 40% less likely to have periodontitis than those who did not engage in any of these health-promoting behaviors [154]. Regarding the dietary content, a study in which subjects consumed four experimental diets (omnivorous higher fat or higher carbohydrate, semi-vegetarian higher fat or higher carbohydrate) found that the semi-vegetarian higher fat group showed improved CAL and decreased gingival crevicular fluid volume after the follow-up period [155]. These studies indicate that lifestyle habits such as an appropriate low-calorie, vegetable-rich diet and moderate exercise can have a positive impact on periodontal tissue health.

Diet has been suggested to possibly influence periodontal disease by altering the oral microflora. In a study by Kato et al., intake of saturated fatty acids and vitamin C were correlated with the diversity of the individual subjects’ bacterial flora (α-diversity), and these nutrients were positively correlated with bacteria such as β-proteobacteria and fusobacteria. Blood glucose load was also positively correlated with the presence of lactobacilli [156]. In recent years, strategies to improve and prevent periodontal disease by actively consuming functional foods and methods to suppress harmful bacteria and promote the growth of beneficial bacteria with foods based on the idea of prebiotics have become a reality [157,158].

Lifestyle modification can be a viable option for the personalization of periodontal care, so dental professionals must lead patients to commit to these modifications using psychological or other techniques. Recently, information and communication technologies, such as mobile device applications, have increasingly been used to support patients in improving their oral health [159].

### 4.4. Low Level Laser Application in Periodontics

The application of lasers in the treatment of periodontal disease has presented a variety of advantages. Laser treatment is promoted for patients who have difficulty with bleeding on surgical treatment or with use of anesthesia. Laser treatment of periodontal disease is classified into high power laser treatment (HPLT) and low-level laser treatment (LLLT), depending on the power of the laser used [160]. HPLT is mainly used for the removal of calculus and inflammatory substances on the root surface, the removal of granulation tissue during periodontal surgery, hemostasis, and gingivectomy. LLLT is expected to be effective for photobiomodulation (PBM). When weak laser energy is used to irradiate a lesion or surgical wound, it affects the immune and metabolic systems, reducing inflammation and promoting wound healing. In recent years, antimicrobial photodynamic therapy (aPDT) has been developed as a new approach, and laser therapy may be an effective option in the precision medicine of periodontal disease, where treatment systems are adapted according to the individual patient’s background.

PBM-based LLLT (PBM therapy, PBMT) has been applied in many medical fields since its clinical efficacy first attracted attention 50 years ago, and its mechanisms have been investigated [161,162]. A meta-analysis by Ren et al. found a significant improvement in PPD and an increase in IL-1β in gingival crevicular fluid when scaling and root planing (SRP) was combined with PBMT compared with SRP alone in the short term [163]. In addition, a meta-analysis of pain relief found that nonsurgical or surgical treatment plus diode laser PBMT provided significant pain relief at 2–7 days compared with no PBMT, and significant postoperative pain relief was observed after adjunctive HPLT using an erbium laser [164].

However, aPDT, which is also based on LLLT, is a therapy that kills nearby bacteria by generating reactive oxygen species (singlet oxygen) through the photocatalytic action of a photosensitizer injected into the lesion site [165]. Methylene blue, toluidine blue, and indocyanine green are mainly used as photosensitizers. The wavelength of the light source to excite the photosensitizer is often 660 nm for methylene blue and toluidine blue and 805 or 810 nm for indocyanine green. These dyes are commercialized in systems that are combined with light source equipment. Because aPDT is a bactericidal therapy, it does not produce physical effects such as removal of calculus under the gingival margin. Therefore, the clinical application of the aPDT concept to the treatment of periodontal disease has long been considered in combination with mechanical debridement [166,167]. More recently, much data have accumulated, and the benefits of aPDT have been reported [168,169]. However, the results of meta-analyses in recent years remain controversial.

In a meta-analysis of SRP alone versus aPDT as an adjunct to SRP, later treatment showed a significant PPD reduction (0.37 mm) and CAL increase (0.33 mm) in chronic periodontitis. This moderate effect was obtained over short durations (after 3 and 6 months) [170]. This improvement of less than 1 mm is consistent with the results of other meta-analyses [171]. However, some studies show no additional benefit of aPDT, possibly due to inconsistencies in study design and other factors [172,173]. The results were similarly controversial if the target periodontal patients were smokers or diabetics [174,175]. Meta-analyses with new computational models may provide new understandings; Ramanauskaite et al. performed a network meta-analysis using the Bayesian model and reported that BOP and PPD decreased significantly when aPDT was performed after subgingival debridement in patients undergoing maintenance [176].

Regarding the comparison between SRP with adjunctive antimicrobial agents and aPDT, several studies indicated that aPDT was more effective than SRP alone in improving PPD and CAL, while antimicrobial agents did not show any significant benefit [177,178]. Conversely, some studies found that antimicrobial agents produced significant improvement, while other adjunctive methods did not [179], so it is unclear whether aPDT is more effective than antimicrobial agents. However, it is possible that aPDT could be an option to avoid overreliance on antimicrobials if it is considered to be equally effective.

Recent trends have reported attempts to develop nanoparticles of photosensitive materials for more efficient drug delivery [180]. Methylene blue-loaded polylactic-co-glycolic acid (PLGA) nanoparticles [181], curcumin-silica nanoparticles [182], copper sulfide nanoparticles together with indocyanine green (ICG) [183], chlorin e6 and coumarin6 in Fe3O4 nanoparticles [184], and many other forms of nanoparticles have been developed. Our group also conducted basic experiments by creating ICG-encapsulated nanospheres. By coating the surface with chitosan, the nanospheres were positively charged, and their adhesion to bacteria was enhanced [185]. Additionally, because the 810 nm diode laser for ICG excitation has tissue permeability, we devised a method of irradiating the laser from outside the periodontal pocket, which is different from the conventional method of inserting the irradiation tip into the pocket [186].

In short, aPDT is still in the process of evolution, and it may become standard in combination treatment with nonsurgical periodontal therapy in the future.

## 5. Future Tasks

To carry out precision medicine in periodontal disease treatment, we are currently examining the true usefulness of a great quantity of information. By building a database of genetic information, biomarkers, and environmental factors that may be related to the disease, and by clarifying the degree of association, the groundwork for precision medicine will be laid. It is important to improve the accuracy of the database by applying deep learning and AI, which are already in use today. The accurate analysis of vast amounts of information is required, and researchers around the world need to continue to work on this with concerted effort. For this reason, it is important that a number of projects related to data collection and database construction be developed in the future. Then, patient stratification should be conducted by classifying patients into several subgroups, so that the pattern and risk of periodontal disease progression can be predicted; this prediction depends largely on oral bacteriological factors and the congenital and acquired biological background and lifestyle of the patient. Patient stratification allows a system to be constructed to select appropriate treatments for individual patients (personalized medicine) based on these predictions of a patient’s susceptibility to disease and the efficacy of treatment based on analysis of this information. In the future, medical practice guidelines should be issued for each individual patient. Appropriate treatment will not be achieved by existing therapies alone but will be created through the development of promising approaches such as regenerative engineering, approaches for systemic disorders, lifestyle and behavioral approaches, and laser application for periodontitis. In regenerative engineering, additive manufacturing has given rise to a new manufacturing concept called four-dimensional (4D) printing [187]. Time is the fourth dimension of the construct of 4D printing, and the goal is to create scaffolds fabricated with smart materials that can respond dynamically to biological conditions or to external stimuli such as pH, humidity, light, and temperature [188,189]. If environmental stimuli can be used to induce appropriate release patterns of angiogenic and osteogenic factors during wound healing and tissue formation, and to enhance regenerative capacity, it is likely that even more ideal periodontal tissue regeneration can be achieved. Additionally, the evolution of treatments for systemic diseases that have been suggested to be associated with periodontal disease should be considered as potentially linked to the evolution of periodontal treatment, and periodontists should pay close attention to this point. Furthermore, dentists need to use psychological techniques and communication technology to strategically encourage lifestyle modification and behavior change as part of personalized periodontal treatment. There is still much that is unknown about the use of laser technology in periodontal treatment, and the elucidation of the detailed mechanisms of action, along with promising therapies such as aPDT, will lead to the development of truly effective lasers. AI will be needed to select highly predictive treatments using the more detailed and voluminous data that will be collected. AI will use not only systematic reviews with a high level of evidence but also a more varied database that includes individual case reports to select the best treatment for each individual patient.

## Figures and Tables

**Figure 1 jpm-12-01743-f001:**
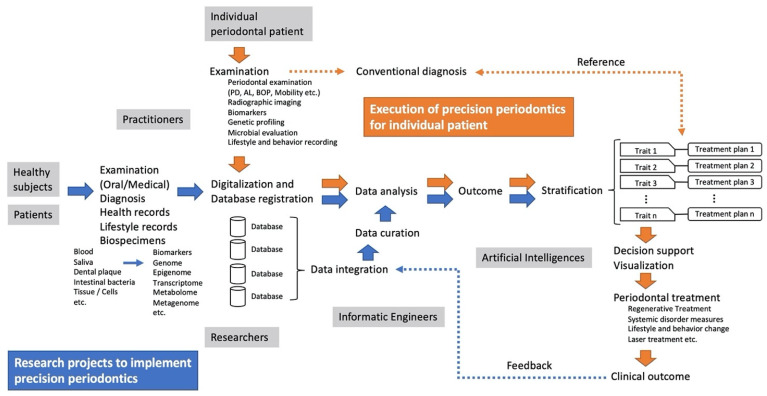
T Conceptual flow chart of precision periodontics. Precision periodontics is the future of periodontal therapy. To realize this vision, it is first necessary to promote the numerous research projects that are currently being conducted in various fields and then to stratify the disease based on the results of these projects. The stratification will be based on traits that are not restricted to the current periodontal disease classification. Finally, each trait needs to be linked to the appropriate treatment (blue flow). However, when precision periodontics is actually executed on an individual patient, data necessary for stratification are first collected from the patient through examinations and are analyzed in conjunction with existing data in the database to determine into which trait category the patient falls. Then, the appropriate treatment for that trait is proposed, and the patient is treated accordingly. The process is automated by artificial intelligence, which supports the decision-making of the clinician (orange flow). The outcomes of the treatment can be fed back to the database to help update the curation of data.

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
