# Peer review of "Next-Generation Examination, Diagnosis, and Personalized Medicine in Periodontal Disease"

_jpm, 2022, doi:10.3390/jpm12101743_

Round 1

Reviewer 1 Report

This manuscript focuses on the fact that the treatment of periodontal disease varies with the condition of each individual patient, and notes that current guidelines for periodontal disease treatment methods are not optimized for the individual. Authors further state that by incorporating the new medical concepts of "precision medicine" and "personalized medicine" into periodontal treatment, it is possible to provide treatment that is more predictive than conventional methods. As a means to achieve this, authors argue that a new diagnostic system that integrates information on individual patient background (biomarkers, genetics, environment, and lifestyle) with conventional medical examination information is needed. As evidence for this, authors cite ongoing studies of new test indicators, including various biomarkers, as well as research on periodontal disease-related genes and the complexity of oral bacteria. I believe that this manuscript is appropriate for publication in this journal because it demonstrates that periodontal disease in the dental and oral fields is one in which the efficacy of "precision medicine" and "personalized medicine" can be fulfilled.

Author Response

Answer to reviewer 1

Your review is greatly appreciated. We also appreciate your generosity and support for the content of this paper. We have made some corrections and ask that you please review them.

We have corrected some wording in Line 527 that was not clear.

Before correction

A meta-analysis of SRP alone versus SRP as an adjunct to aPDT showed a significant PPD reduction (0.37 mm) and CAL increase (0.33 mm) in the treatment of chronic periodontitis.

After correction

In a meta-analysis of SRP alone versus aPDT as an adjunct to SRP, later treatment showed a significant PPD reduction (0.37 mm) and CAL increase (0.33 mm) in chronic periodontitis.

For English spelling, we will have the editorial team check it.

Reviewer 2 Report

Dear Authors, I want to congratulate you because your review article seems to me extremely well done and complete in the evaluation of the diagnosis and treatment of periodontal disease.

The only note concerns the sentence "A meta-analysis of SRP alone ..." in line 573, where it is not clear which of the two treatments obtains the best results.

For the rest I believe that your article certainly deserves to be published on JPM

Best regards

Author Response

Answer to reviewer 2

Your review is greatly appreciated. We also appreciate your generosity and support for the content of this paper. We have made the following corrections to the points you raised. We ask that you please review it.

Before correction

A meta-analysis of SRP alone versus SRP as an adjunct to aPDT showed a significant PPD reduction (0.37 mm) and CAL increase (0.33 mm) in the treatment of chronic periodontitis.

After correction

In a meta-analysis of SRP alone versus aPDT as an adjunct to SRP, later treatment showed a significant PPD reduction (0.37 mm) and CAL increase (0.33 mm) in chronic periodontitis.

For English spelling, we will have the editorial team check it.
